# Sphingosine 1-Phosphate and Apolipoprotein M Levels and Their Correlations with Inflammatory Biomarkers in Patients with Untreated Familial Hypercholesterolemia

**DOI:** 10.3390/ijms232214065

**Published:** 2022-11-15

**Authors:** Lilla Juhász, Hajnalka Lőrincz, Anita Szentpéteri, Bíborka Nádró, Éva Varga, György Paragh, Mariann Harangi

**Affiliations:** 1Division of Metabolic Diseases, Department of Internal Medicine, Faculty of Medicine, University of Debrecen, 4032 Debrecen, Hungary; 2Doctoral School of Health Sciences, Faculty of Public Health, University of Debrecen, 4032 Debrecen, Hungary; 3Department of Internal Medicine and Hematology, Semmelweis University, 1085 Budapest, Hungary; 4ELKH-UD Vascular Pathophysiology Research Group 11003, University of Debrecen, 4032 Debrecen, Hungary

**Keywords:** sphingosine 1-phosphate, apolipoprotein M, familial hypercholesterolemia, sCD40 ligand, soluble vascular adhesion molecule-1, arylesterase activity, high-density lipoprotein, inflammation

## Abstract

High-density lipoprotein (HDL)-bound apolipoprotein M/sphingosine 1-phosphate (ApoM/S1P) complex in cardiovascular diseases serves as a bridge between HDL and endothelial cells, maintaining a healthy endothelial barrier. To date, S1P and ApoM in patients with untreated heterozygous familial hypercholesterolemia (HeFH) have not been extensively studied. Eighty-one untreated patients with HeFH and 32 healthy control subjects were included in this study. Serum S1P, ApoM, sCD40L, sICAM-1, sVCAM-1, oxLDL, and TNFα concentrations were determined by ELISA. PON1 activities were measured spectrophotometrically. Lipoprotein subfractions were detected by Lipoprint. We diagnosed FH using the Dutch Lipid Clinic Network criteria. Significantly higher serum S1P and ApoM levels were found in HeFH patients compared to controls. S1P negatively correlated with large HDL and positively with small HDL subfractions in HeFH patients and the whole study population. S1P showed significant positive correlations with sCD40L and MMP-9 levels and PON1 arylesterase activity, while we found significant negative correlation between sVCAM-1 and S1P in HeFH patients. A backward stepwise multiple regression analysis showed that the best predictors of serum S1P were large HDL subfraction and arylesterase activity. Higher S1P and ApoM levels and their correlations with HDL subfractions and inflammatory markers in HeFH patients implied their possible role in endothelial protection.

## 1. Introduction

Heterozygous familial hypercholesterolemia (HeFH) is the most frequent metabolic monogenic disorder with high risk for premature cardiovascular disease (CVD). The estimated prevalence of HeFH in European countries is 1:340 [1]. However, patients with the heterozygous genotype show wide phenotypic heterogeneity with extremely variable prevalence of CVD even in patients sharing the same mutations [2]. Although accelerated atherogenesis in HeFH is mostly driven by markedly increased low-density lipoprotein (LDL) cholesterol levels, previous studies showed that besides non-specific and specific risk factors [3], low HDL cholesterol (HDL-C) may contribute to the different incidence and severity of chronic heart disease (CHD) in the HeFH population [4]. Moreover, qualitative abnormalities of HDL including functional and structural changes were also reported in HeFH [5,6,7]. Human paraoxonase-1 (PON1) is a calcium-dependent esterase with beneficial anti-atherogenic properties, and its activity is greater in HDL3 particles [8]. PON1 status was found to be altered in HeFH [9] and significantly improved during statin treatment [10]. In addition, Versmissen et al., found lower plasma efflux capacity in HeFH patients with CVD compared to those without CVD. Based on their results, increased sphingosine 1-phosphate (S1P) and apolipoprotein M (ApoM) concentrations in HDL might play a role in HDL functionality including efflux capacity [11].

Bioactive S1P is a lipid mediator exported out of cells, mainly red blood cells (RBCs), platelets, fibroblasts, vascular smooth muscle cells (VCMCs), endothelial cells (ECs), and cardiomyocytes, as part of the so-called “inside-out” signaling [12]. In addition, extracellular sphingosine kinase 1 (SphK1) is released from these cells, participating in S1P generation [13]. S1P’s effect is transmitted by specific G protein-coupled receptors (GPCRs), called S1P receptors (S1PRs). Five S1PRs (S1PR1–5) have been identified so far, differing in tissue and cell expression, with S1PR1–3 primarily expressed in the cardiovascular system, in ECs and cardiomyocytes, out of which S1PR1 is the most expressed S1PR subtype [14]. Through the activation of S1PRs, S1P plays a pivotal role in the development and stabilization of the vasculature [15]. Circulating S1P is situated in the highest concentration in small-sized HDL3 particles and bound with high affinity to ApoM in HDL [16], with only low affinity to albumin.

A previous study examined the constituents of HDL including S1P and ApoM in 13 male temporarily untreated FH patients with and without CHD and their non-FH brothers. Compared to their non-FH brothers, FH patients without CHD displayed significantly higher serum HDL-C, HDL-S1P, and ApoM, while FH patients with CHD displayed lower concentrations than their non-FH brothers [11]. In a case-control study, the HDL-S1P content and the antioxidant capacity of HDL were measured in 12 FH patients with and without statin treatment and in 12 healthy controls. HDL-associated S1P significantly correlated with cell protection but not with HDL-C or ApoA1. Neither HDL’s S1P content nor HDL’s protective capacity differed between nontreated FH patients and controls. Moreover, statin treatment had no effect on any of these parameters [17]. However, serum levels of S1P and ApoM have not been investigated in larger FH populations and their correlations with inflammatory, endothelial, and oxidative serum markers have not been explored. 

Therefore, we aimed to detect serum S1P, ApoM, soluble intercellular adhesion molecule-1 (sICAM-1), soluble vascular cell adhesion molecule-1 (sVCAM-1), the soluble form of CD40 ligand (sCD40L), oxidized LDL (oxLDL), myeloperoxidase (MPO), high-sensitivity C-reactive protein (hsCRP), and LDL and HDL subfractions as well as PON1 paraoxonase and arylesterase activities in a large, untreated HeFH population. We hypothesized that serum HDL-C, S1P, and ApoM levels would be higher in untreated patients with HeFH and circulating S1P would correlate with the concentrations of inflammatory, oxidative, and lipoprotein parameters.

## 2. Results

Compared to controls, HeFH patients had significantly higher total cholesterol, LDL-C, triglyceride, apoB100, and Lp(a) levels, while circulating HDL-C and ApoA1 did not differ significantly. Significantly higher PON1 arylesterase activity was found in HeFH patients compared to controls, but there were no differences in PON1 paraoxonase and salt-stimulated paraoxonase activities between the two study groups. Circulating oxLDL, myeloperoxidase, sICAM-1, and TNFα were significantly higher in HeFH patients than in non-FH subjects. However, there were no significant differences in sVCAM-1, sCD40L, and hsCRP between patients and controls. Moreover, ApoM and S1P levels were significantly higher in HeFH patients compared to controls (Table 1, Figure 1a,b). 

There were no significant differences in ApoM and S1P concentrations between HeFH patients with (VC) or without vascular complications (nonVC), including previous acute myocardial infarction, stroke, carotid artery atherosclerotic disease, or peripheral arterial disease (ApoM: VC 3.79 ± 0.62 vs. nonVC 3.73 ± 0.53 μg/mL, *p* = 0.7; S1P: VC 7.63 ± 1.08 vs. nonVC 7.75 ± 2.06 ng/mL, *p* = 0.8, respectively). We could not find significant differences in the S1P/ApoM ratio between FH patients and controls (2120 ± 700 vs. 2290 ± 740; *p* = 0.22).

The proportion and absolute amount of VLDL and IDL subfractions were significantly higher in HeFH patients compared to control individuals. Both proportions and levels of large- and small-density LDL subfractions were significantly higher while mean LDL size was significantly lower in HeFH patients compared to controls. Furthermore, a lower percentage and concentration of large and intermediate HDL subfractions, in contrast to a higher percentage and concentration of small HDL subfractions, were found in HeFH patients compared to control subjects (Table 2).

We found significant negative correlations between large HDL subfractions and S1P levels in the whole study population and in HeFH patients (Figure 2a,d). There were no correlations between intermediate HDL and S1P levels in any study populations (Figure 2b,e). In addition, there were significant positive correlations between small HDL subfractions and S1P concentrations both in the whole study population and in HeFH patients (Figure 2c,f).

Significant negative correlation was found between sVCAM-1 and S1P in the HeFH population (Figure 3a). We found significant positive correlations among PON1 arylesterase activity, sCD40L, MMP-9, and S1P concentrations (Figure 3b–d). The S1P/ApoM ratio showed significant negative correlation with sVCAM-1 (r = −0.197; *p* = 0.037), while there was a positive correlation between S1P/ApoM and sCD40L (r = 0.28; *p* = 0.0027) (data not shown).

Multiple regression analysis using a backward stepwise method showed that serum S1P’s best predictors turned out to be PON1 arylesterase activity (β = 0.281; *p* < 0.001) and the percentage of large HDL subfractions (β = 0.35; *p* < 0.01). The model contained logTG, a percentage of large HDL, a percentage of small HDL, sVCAM-1, ApoM, PON1 arylesterase activity, MMP-9, and logCD40L (Table 3).

## 3. Discussion

Despite clear epidemiologic evidence that low serum HDL-C is a risk factor for CVD, no clinical trial aiming at raising HDL-C has been successful in reducing risk. Thus, the concept has been introduced that HDL function, rather than HDL-C concentration is responsible for the beneficial effects and that improving function is the true therapeutic target since HDL has numerous potentially atheroprotective functions [18,19,20,21]. However, the clarification of regulatory pathways and identification of biomarkers responsible for these anti-atherogenic characteristics is essential to step forward. Indeed, HDL-associated S1P has been shown to causally contribute to many HDL functions, such as the maintenance of endothelial homeostasis, arterial vasodilation, and cardioprotection [22,23]. However, most of those studies examined HDL from healthy patients, leaving the contribution of disease-associated alterations of HDL-S1P to numerous aspects of HDL dysfunction unexplored [24]. To date, this is the first clinical study evaluating S1P levels, HDL subfraction distribution, HDL function, and inflammatory markers in HeFH. In contrast to previous data on smaller HeFH groups, we found higher S1P and ApoM levels in our HeFH population. However, opposite to this previous study [11], S1P and ApoM concentrations were similar in HeFH patients with and without vascular complications. To date, S1P and ApoM concentrations have not been determined in larger, unrelated FH patient populations.

Previous prospective studies showed that several molecular proinflammatory biomarkers from foam cell formation to plaque rupture may be used to predict future cardiovascular events. While the elevation of serum inflammatory cytokines and adhesion molecules such as sVCAM-1 and sICAM-1 can be detected in the early phase of atherogenesis, increased levels of oxLDL, MMPs, and sCD40L can be seen in the late phase, indicating plaque destabilization and imminent rupture [25]. In our HeFH patients, S1P concentration correlated negatively with sVCAM-1, but there was a positive correlation between sCD40L and MMP-9 concentrations. It was reported that S1P transported by HDL-associated ApoM may act on S1P1 and three receptors, inducing anti-atherogenic and vasculoprotective effects [26], while S1P carried by HDL-ApoM can also bind to S1PR2, leading to macrophage retention in the atherosclerotic plaques and the promotion of atherosclerosis [27]. Therefore, in our HeFH patients, the effect of higher ApoM and S1P concentrations might not necessarily be beneficial. Keul et al., proved that HDL-bound S1P exerts a potent anti-inflammatory effect on smooth muscle cells by inhibiting the induction of TNFα-stimulated inflammatory genes, including MMP-9 [28]. However, we found a positive correlation between S1P and MMP-9 concentrations, which may indicate the responsive expression of S1P in mature atherosclerotic plaques.

The function of HDL-associated enzymes is often impaired in HeFH patients [4]. A key role in the antioxidant properties of HDL is exerted by the enzyme PON1 associated to the HDL surface [29]. PON1 hydrolyzes oxidized lipids and protects LDL and biological membranes from lipid peroxidation, resulting in decreased endogenous oxidative stress and the prevention of atherogenesis [30]. Previously, decreased PON1 arylesterase activity was reported in FH patients [9]. In contrast, MPO is a pro-oxidant enzyme produced mainly in neutrophils and monocytes; it generates reactive intermediates. In the present study, we unexpectedly found higher PON1 arylesterase activity in HeFH compared to controls, although we enrolled untreated patients to exclude the previously reported effect of statins on PON1 activities [10,31,32]. We could not find significant differences between the PON1 paraoxonase and salt-stimulated paraoxonase activities of HeFH patients and the controls, while the higher oxLDL and increased MPO activity demonstrated increased oxidative stress in our HeFH population, which was similar to some other previous observations. Of note, PON1 arylesterase activity was found to be a predictive factor of S1P based on the result of multiple regression analysis, indicating the link between S1P and HDL-associated antioxidative processes. The lower percentage and concentration of large and intermediate HDL subfractions, in contrast to a higher percentage and concentration of small HDL subfractions, in HeFH patients compared to control subjects were described previously [33]. Recently, higher concentrations of large HDL particles were found in CHD-free elderly HeFH patients, potentially indicating that these particles have other functions than smaller HDLs and that separation of HDL subfractions might provide better risk profiles in HeFH than the currently generally used HDL-C concentration [34].

Some limitations of the study must be mentioned. The direct association of S1P with HDL particles was not measured. HDL-S1P could be determined by liquid chromatography–mass spectrometry. However, this time-consuming and costly method is not available in the everyday clinical practice; therefore, HDL-S1P could not be used as a biomarker. The use of imaging modalities to identify and quantify the burden of atherosclerosis in the aorta, carotid arteries, coronary arteries, and peripheral vasculature would improve the value of the study. However, the results underline the potential importance of studying HDL function and the potential regulatory role of S1P in HeFH.

## 4. Materials and Methods

### 4.1. Study Population

Eighty-one subjects (55 females and 26 males) with HeFH were enrolled in our study at the Lipid Outpatient Clinic of the Department of Internal Medicine, University of Debrecen. All HeFH patients were heterozygous with a confirmed LDL receptor gene mutation or fulfilled the Dutch Lipid Clinic Network diagnostic criteria for FH [35]. The patients were referred to our Lipid Outpatient Clinic by general practitioners, cardiologists, and neurologists to verify the diagnosis of HeFH and initiate optimal therapy. We asked the patients to arrive after 12 h of fasting from 08:00–10:00 a.m. All HeFH patients were newly diagnosed without ongoing lipid-lowering drug treatment. Thirty-two gender- and aged-matched healthy individuals at our General Outpatient Clinic Department of Internal Medicine, University of Debrecen, were used as controls. In controls, the main inclusion criteria were normal body mass index; normal cholesterol, glucose, and liver enzyme levels; being free of medications; and no previous chronic or acute diseases in the past 3 months. A physical examination and electrocardiogram did not show any abnormalities.

Participants with a type 1 or 2 diabetes mellitus were excluded. Other exclusion criteria were alcoholism; pregnancy; lactation; malignancy; known liver, autoimmune, and endocrine diseases; and chronic neurological and hematological disorders, which can be associated with peripheral neuropathy. All participants provided written, informed consent before enrollment. The study was carried out in accordance with the Declaration of Helsinki and was approved by the local and regional ethical committees (DE RKEB/IKEB 4775-2017, date of approval: 3 April 2020, and ETT TUKEB 34952-1/2017/EKU, date of approval: 30 June 2017).

### 4.2. Blood Sampling

Venous blood samples were drawn in the morning into Vacutainer tubes and sera were centrifuged immediately at 3500 RPM for 15 min. Triglyceride, total cholesterol, low-density lipoprotein cholesterol (LDL-C), high-density lipoprotein cholesterol (HDL-C), lipoprotein(a) (Lp(a)), creatinine, uric acid, and glucose were analyzed from fresh samples with a Cobas c600 autoanalyzer (Roche Ltd., Mannheim, Germany) in the Department of Laboratory Medicine, Faculty of Medicine, University of Debrecen, Hungary. Tests were performed according to the manufacturer’s recommendations, and reagents were purchased from the same vendor. The sera for subsequent enzyme-linked immunoassays (ELISAs), enzyme activity measurements, and lipoprotein subfraction analyses were kept at −70 °C before analysis.

### 4.3. Measurement of ApoM and S1P

Serum ApoM levels were determined with an ELISA kit (BioVendor—Laboratorni medicina a.s., Brno, Czech Republic) and expressed as µg/mL. Intra-assay coefficients of variation ranged from 4.9–5.22% and inter-assay coefficients of variation ranged from 5.7–5.8%. S1P was also measured by ELISA (Echelon Biosciences, Salt Lake City, UT, USA), according to the instructions of the manufacturer, and expressed as μg/mL.

### 4.4. TNF-α Measurement

Serum high-sensitivity TNF-α levels were measured using Quantikine TNF-α ELISA (R&D Systems Europe Ltd., Abington, UK) according to the recommendations of the manufacturer; values were expressed as pg/mL. Intra-assay coefficients of variation ranged from 1.9 to 2.2% and inter-assay coefficients of variation ranged from 6.2 to 6.7%.

### 4.5. Oxidized LDL Measurement

Serum concentrations of oxidized LDL (oxLDL) were detected by commercial sandwich ELISA (Mercodia AB, Uppsala, Sweden) based on a direct sandwich technique where two monoclonal antibodies target separate antigenic determinants of the oxidized apolipoprotein B molecule. Sensitivity of oxLDL measurements was <1 mU/L and intra- and inter-assay coefficients of variation were 5.5–7.3% and 4–6.2%, respectively.

### 4.6. The sICAM-1, sVCAM-1, and sCD40L Measurements

Serum sICAM-1, sVCAM-1, and sCD40L were assessed with sandwich ELISAs (R&D Systems Europe Ltd., Abington, UK). ELISA procedures were carried out according to the instructions of the manufacturer. Intra-assay coefficients of variations were 3.7–5.2% (sICAM-1), 2.3–3.6% (sVCAM-1), and 4.5–5.4% (sCD40L), while the inter-assay coefficients of variation ranged between 4.4 and 6.7% (sICAM-1), 5.5 and 7.8% (sVCAM-1), and 6.0 and 6.4% (sCD40L). All concentration values were expressed as ng/mL.

### 4.7. Myeloperoxidase Measurement

Serum concentrations of MPO were measured by ELISA (R&D Systems Europe Ltd., Abington, UK), and the assay was performed according to the instructions of the manufacturer. Intra- and inter-assay coefficients of variation were 6.5–9.4%.

### 4.8. Determination of PON1 Enzyme Activities

Serum PON1 paraoxonase activity was monitored by a kinetic, semiautomated method using paraoxon (O,Odiethyl-O-p-nitrophenyl phosphate, Sigma-Aldrich, Budapest, Hungary) as a substrate. Hydrolysis of paraoxon was followed at 405 nm at +22–24 °C. Serum PON1 arylesterase activity was measured using phenylacetate as substrate (Sigma Aldrich, Budapest, Hungary), and the hydrolysis of the substrate was monitored at 270 nm at +22–24 °C, as previously described [36].

### 4.9. Determinations of Lipoprotein Subfractions

LDL and HDL lipoprotein subfractions were distributed based on their size by Lipoprint System (Quantimetrix Corporation, Redondo Beach, CA, USA), as previously described [37,38]. Briefly, 25 µL sera were taken into polyacrylamide gel tubes with 200 and 300 µL loading gel containing Sudan Black, respectively. After 30 min of photopolymerization, the gel tubes were electrophorized in an electrophoresis chamber with 3 mA/each tube. Each electrophoresis was loaded with a high purity lipoprotein quality control, which was provided by Quantimetrix (Liposure Serum Lipoprotein Control, Quantimetrix Corp., Redondo Beach, CA, USA). After a half hour but no longer than 2 h of rest, the lipoprotein bands were scanned with an ArtixScan M1 digital scanner (Microtek International Inc., Hsinchu, Taiwan) and analyzed with the Lipoware Software developed by the manufacturer (Quantimetrix Corp., Redondo Beach, CA, USA).

In the case of the LDL subfraction analysis, up to seven LDL subfractions were determined between the VLDL and HDL peaks. The proportion of large LDL (large LDL %) was defined as the summed percentages of LDL1 and LDL2, whereas the proportion of the small LDL (small-dense LDL %) was defined as the sum of LDL3–LDL7. Cholesterol concentrations of the LDL subfractions were determined by multiplying the relative area under the curve (AUC) of subfractions by the total cholesterol concentration. The calculated total LDL-C was the sum of cholesterol in midbands C through A (which are mainly comprised of IDL) plus LDL subfractions (LDL1–LDL7). The calculated LDL-C correlated with directly measured LDL-C (Lipoprint LDL: 130.8 ± 30.14 mg/dL vs. β-Quant LDL: 130.0 ± 30.42 mg/dL, r^2^ = 0.887), as described previously [20].

In the case of the HDL subfraction analysis, 10 HDL subfractions were determined: large (HDL1–3), intermediate (HDL4–7), and small (HDL8–10) HDL subfractions were distributed between the VLDL + IDL + LDL and albumin bands. The cholesterol content of the HDL subfractions was calculated by the Lipoware Software according to the relative AUC of subfraction bands.

### 4.10. Statistical Analyses

Statistical analyses were performed using the Statistica 13.5.0.17 (TIBCO Software Inc., Tulsa, OK, USA), and graphs were made using the GraphPad Prism 6.01 (GraphPad Prism Software Inc., San Diego, CA, USA). We also calculated the statistical power with the SPH Analytics online calculator (SPH Analytics LTD., Alpharetta, GA, USA) to validate the difference of the circulating ApoM and S1P levels in HeFH (group 1) and control subjects (group 2). The statistical power was above 0.8 (0.98). The difference between the genders in the two studied groups was calculated with the chi-square test. The normality of the continuous variables was tested using the Kolmogorov–Smirnov test. In the case of normal distribution, the comparison between groups was analyzed with the Student’s unpaired *t*-test and with the Mann–Whitney U-test in the case of variables with non-normal distribution, respectively. Data were presented as means ± standard deviation or medians (upper and lower quartiles). The relationship between normally distributed variables was performed with Pearson tests. Backward multiple regression analysis was performed to define which variable(s) is/are the best predictor(s) of S1P levels. The *p* ≤ 0.05 probability values were considered statistically significant.

## 5. Conclusions

This is the first clinical study evaluating S1P concentrations, HDL subfraction distribution, HDL function, and inflammatory markers in HeFH. The effect of higher ApoM and S1P in HeFH might be complex; ApoM/S1P may exert an anti-inflammatory effect on early-phase atherosclerotic plaques characterized by the concentrations of biomarkers such as adhesion molecules including sVCAM-1. However, increased S1P in HeFH patients may induce the expression of late atherosclerotic biomarkers including sCD40L and MMP-9 in complicated lesions. Our results demonstrate the complex regulatory role of the ApoM/S1P complex on vascular function in HeFH patients and highlight the importance of further studies to clarify the consequences of high ApoM and S1P concentrations in heterozygous familial hypercholesterolemia.

## Figures and Tables

**Figure 1 ijms-23-14065-f001:**
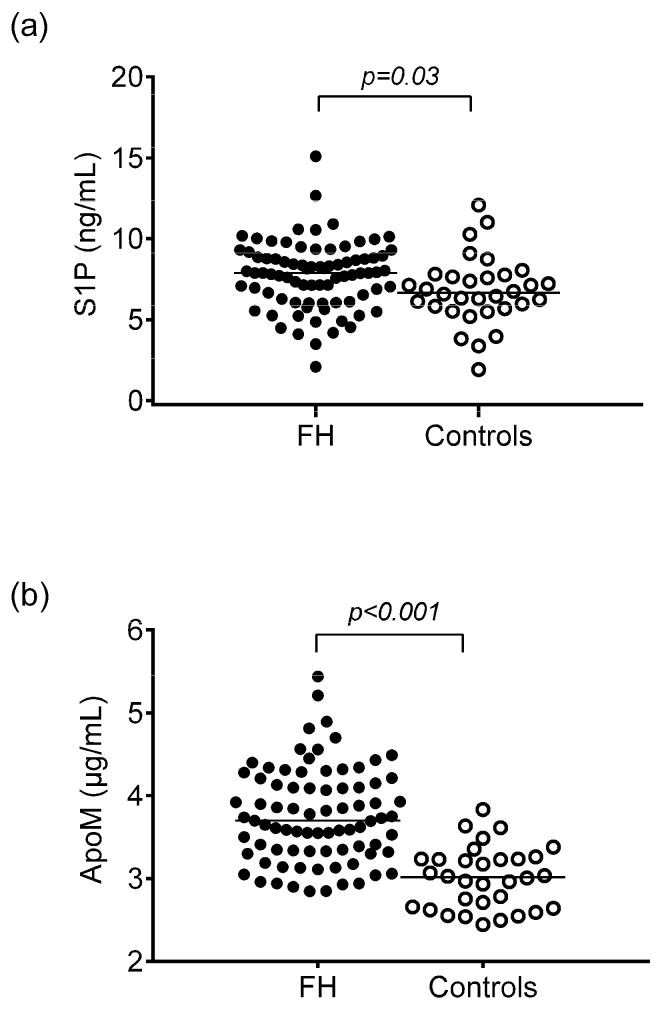
Serum concentrations of sphingosine 1-phosphate (S1P) (**a**) and apolipoprotein M (ApoM) (**b**) in patients with heterozygous familial hypercholesterolemia (HeFH) and controls.

**Figure 2 ijms-23-14065-f002:**
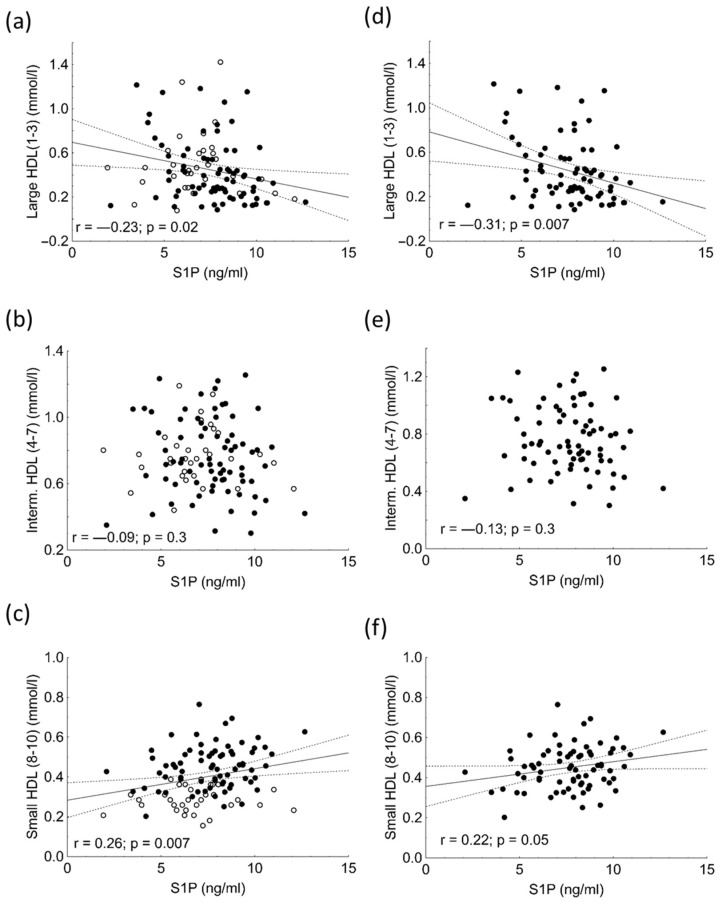
Correlations between the serum concentrations of large HDL subfraction and sphingosine 1-phosphate (S1P) in the in the whole study population (**a**) and in heterozygous familial hypercholesterolemic (HeFH) patients (**d**). Correlations between the serum concentrations of intermediate HDL subfraction and sphingosine 1-phosphate (S1P) in the in the whole study population (**b**) and in heterozygous familial hypercholesterolemic (HeFH) patients (**e**). Correlations between the serum levels of intermediate HDL subfraction and sphingosine 1-phosphate (S1P) in the whole study population (**c**) and in heterozygous familial hypercholesterolemic (HeFH) patients (**f**).

**Figure 3 ijms-23-14065-f003:**
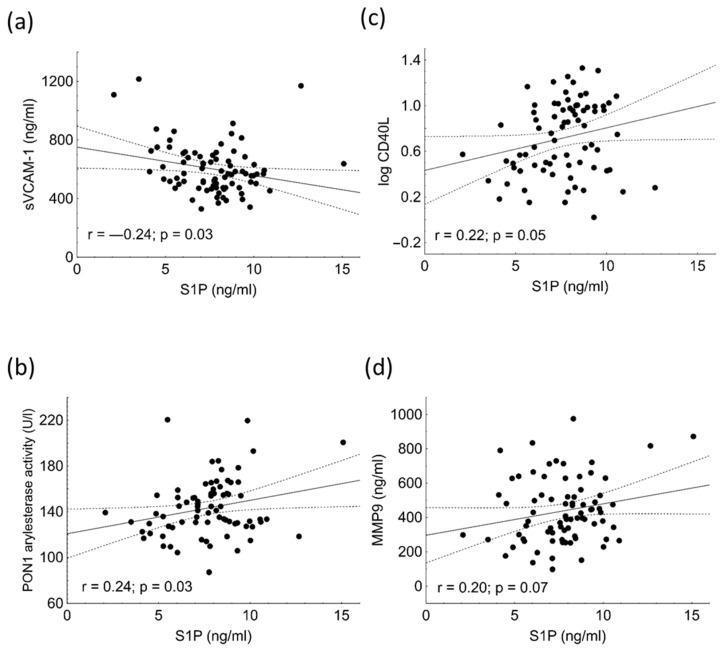
Correlations among sVCAM-1 (**a**), PON1 arylesterase activity (**b**), serum concentrations of logCD40L (**c**), MMP-9 (**d**), and sphingosine 1-phosphate (S1P) in heterozygous familial hypercholesterolemic (HeFH) patients.

**Table 1 ijms-23-14065-t001:** Anthropometric and laboratory parameters of study individuals. Values are presented as mean ± standard deviation or median (lower quartile, upper quartile).

	FH Patients	Controls	*p* Values
Number of subjects	81	32	
Male/female	26/55	5/27	ns.
Age (years)	53.22 ± 14.5	41.8 ± 6.0	*p* < 0.001
Cholesterol (mmol/L)	8.87 ± 1.47	5.07 ± 0.78	*p* < 0.001
HDL-C (mmol/L)	1.62 ± 0.48	1.56 ± 0.46	ns.
LDL-C (mmol/L)	6.48 ± 1.28	2.93 ± 0.52	*p* < 0.001
Triglyceride (mmol/L)	1.6 (1.0–2.4)	1.0 (0.75–1.39)	*p* < 0.001
ApoB100 (g/L)	1.78 ± 0.38	0.94 ± 0.18	*p* < 0.001
ApoA1 (g/L)	1.71 ± 0.28	1.68 ± 0.31	ns.
Lp(a) (mg/L)	179 (75–857)	90 (30–214)	*p* < 0.05
ApoM (μg/mL)	3.76 ± 0.57	3.01 ± 0.37	*p* < 0.001
S1P (ng/mL)	7.73 ± 2.07	6.79 ± 2.09	*p* < 0.05
S1P/ApoM ratio	2120 ± 700	2290 ± 740	ns.
hsCRP (mg/L)	1.84 (0.70–2.90)	1.55 (0.6–2.95)	ns.
PON1 paraoxonase activity (U/L)	107.02 (43.61–166.5)	83.0 (47.9–167.4)	ns.
PON1 salt-stimulated paraoxonase activity (U/L)	183.5 (103.2–322.6)	169.4 (97.3–297.4)	ns.
PON1 arylesterase activity (U/L)	143.2 ± 25.12	135.4 ± 36.8	*p* < 0.01
Myeloperoxidase (ng/mL)	297.7 (158.15–456.5)	135.7 (99.4–195.1)	*p* < 0.001
oxLDL (U/L)	187.98 ± 71.04	41.1 ± 9.57	*p* < 0.001
sICAM-1 (ng/mL)	270.66 ± 69.9	210.8 ± 32,2	*p* < 0.001
sVCAM-1 (ng/mL)	573.9 ± 140.45	467.7 ± 106.3	ns.
sCD40L (ng/mL)	10.02 ± 4.3	8.22 ± 3.44	ns.
TNFα (pg/mL)	0.47 ± 0.17	1.66 ± 0.91	*p* < 0.001

Abbreviations: ApoA1: apolipoprotein A1; ApoB100: apolipoprotein B100; FH: familial hypercholesterolemia; HDL: high-density lipoprotein; hsCRP: high-sensitivity C-reactive protein; LDL: low-density lipoprotein; Lp(a): lipoprotein(a); PON1: paraoxonase-1; sCD40L: soluble CD40 ligand; sICAM-1: soluble intercellular adhesion molecule-1; sVCAM-1: soluble vascular adhesion molecule-1; TNFα: tumor necrosis factor alpha.

**Table 2 ijms-23-14065-t002:** Concentrations and proportions of lipoprotein subfractions in study participants. Values are expressed as mean ± SD or median (lower quartile, upper quartile).

	FH Patients	Controls	*p*
VLDL subfraction (%)	19.76 ± 5.8	16.95 ± 2.2	0.01
VLDL subfraction (mmol/L)	1.77 ± 0.66	0.868 ± 0.17	<0.001
Midband (IDL) (%)	28.89 ± 4.5	29.83 ± 4.9	ns
Midband (IDL) (mmol/L)	2.52 ± 0.62	1.505 ± 0.38	<0.001
LDL subfractions			
Large LDL (%)	27.3 ± 5.5	20.9 ± 5.8	<0.001
Small LDL (%)	3.2 (1.1–11.0)	0.5 (0–0.8)	<0.001
Large LDL (mmol/L)	2.29 (2.05–2.64)	1.047 (0.827–1.344)	<0.001
Small-density LDL (mmol/L)	0.18 (0.05–0.79)	0.026 (0–0.052)	<0.001
Mean LDL size (nm)	26.78 ± 0.58	27.26 ± 0.37	<0.05
HDL subfractions			
Large HDL (%)	24.7 ± 11.0	30.2 ± 8.9	0.02
Intermediate HDL (%)	46.0 ± 4.9	50.6 ± 4.7	<0.001
Small HDL (%)	29.3 ± 10.6	19.2 ± 5.4	<0.001
Large HDL (mmol/L)	0.35 (0.231–0.571)	0.453 (0.31–0.608)	<0.001
Intermediate HDL (mmol/L)	0.72 (0.613–0.932)	0.750 (0.659–0.853)	<0.05
Small HDL (mmol/L)	0.452 (0.374–0.523)	0.284 (0.246–0.336)	<0.01

Abbreviations: HDL: high-density lipoprotein; IDL: intermediate-density lipoprotein; LDL: low-density lipoprotein; VLDL: very-low-density lipoprotein.

**Table 3 ijms-23-14065-t003:** Backward stepwise multiple regression analysis to determine significant predictor(s) of sphingosine 1-phosphate (S1P).

Variable	β	*p*-Value
log triglyceride	0.1	0.5
large HDL (%)	0.35	<0.01
small HDL (%)	0.136	0.8
sVCAM-1	−0.13	0.2
ApoM	−0.1	0.2
PON1 arylesterase activity	0.281	<0.001
MMP-9	0.142	0.3
log sCD40L	−0.16	0.7

Abbreviations: ApoM: apolipoprotein M; HDL: high-density lipoprotein; MMP-9: matrix metalloproteinase-9; PON1: human paraoxonase-1; sCD40L: soluble CD40 ligand; sVCAM-1: soluble vascular cell adhesion molecule-1.

## Data Availability

All data generated or analyzed during this study are included in this published article. All data generated or analyzed during the current study are available from the corresponding author on reasonable request.

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
