# Peer review of "Sphingosine 1-Phosphate and Apolipoprotein M Levels and Their Correlations with Inflammatory Biomarkers in Patients with Untreated Familial Hypercholesterolemia"

_ijms, 2022, doi:10.3390/ijms232214065_

Round 1
Reviewer 1 Report
This study describes the correlation of ApoM/S1P with other biomarkers in HeFH. The authors have presented clear data. However, there are also data of laboratory parameters such as sICAM-1 and TNF-a which appear to be significant but was not included in the correlation analysis.
Author Response
Dear Reviewer, thank you for your positive reply and helpful comments on our manuscript. We hereby answer your recommendations as follows:
- This study describes the correlation of ApoM/S1P with other biomarkers in HeFH. The authors have presented clear data. However, there are also data of laboratory parameters such as sICAM-1 and TNF-a which appear to be significant but was not included in the correlation analysis.
Response: Thank you for the comment. You are right, there were significant differences in sICAM-1 and TNF-α levels between HeFH and control subjects. Therefore, we investigated the correlations between sICAM-1 and S1P, and between TNF-α and S1P levels, but these correlations were not significant in the whole study population (sICAM: r=0.18; p=0.0579; TNF- α: r=-0.17; p=0.077) and in HeFH patients (sICAM: r=0.11; p=0.3; TNF- α: r=0.0047; p=0.97). Therefore, these parameters are not predictors, and were not included into the backward stepwise multiple regression analysis.
Again, we are very thankful for your valuable and thorough review.
Reviewer 2 Report
1. This is an interesting study, the authors used the appropriated clinical samples in this study, especially the manuscript mentioned that “all patients were newly diagnosed without ongoing lipid-lowering drug treatment”, this was an important base of this study.
2. The results of this study are convincing. I just want to suggest the authors to discuss 3 questions:
(1) Since High-density lipoprotein (HDL)-bound apolipoprotein M/sphingosine 1-phosphate (ApoM/S1P) complex in cardiovascular diseases serves as a bridge between HDL and endothelial cells, maintaining a healthy endothelial barrier. How to distinguish the results that showed in Figure 1, Serum concentrations of S1P and ApoM from patients were single molecular of ApoM or S1P or their complex form.
(2) The level of ApoM is ug/ml and the level of S1P is ng/ml, how about presenting the results with Molar concentration, it may help readers to realize the (ApoM/S1P) complex.
(3) How about the role of Lysophosphatidic acid (LPA) in heterozygous familial hypercholesterolemia.
Author Response
Dear Reviewer, thank you for your helpful comments and suggestions on our manuscript. We hereby answer your recommendations as follows:
This is an interesting study, the authors used the appropriated clinical samples in this study, especially the manuscript mentioned that “all patients were newly diagnosed without ongoing lipid-lowering drug treatment”, this was an important base of this study.
The results of this study are convincing. I just want to suggest the authors to discuss 3 questions:
- Since High-density lipoprotein (HDL)-bound apolipoprotein M/sphingosine 1-phosphate (ApoM/S1P) complex in cardiovascular diseases serves as a bridge between HDL and endothelial cells, maintaining a healthy endothelial barrier. How to distinguish the results that showed in Figure 1, Serum concentrations of S1P and ApoM from patients were single molecular of ApoM or S1P or their complex form.
Response:
Thank you for the comment. As we noted among the limitations of the study, direct associations of S1P to HDL particles (consequently, association of S1P to ApoM) were not measured. HDL-S1P could be determined by liquid-chromatograpy-mass spectrometry, but this time consuming and costly method is not available in the everyday clinical practice, therefore, it could not be used as a biomarker. Still, our results indicate its complex role in atherogenesis in HeFH patients. Further studies are needed to clarify the role of complex and free forms of these molecules.
- The level of ApoM is ug/ml and the level of S1P is ng/ml, how about presenting the results with Molar concentration, it may help readers to realize the (ApoM/S1P) complex.
Response: Thank you for the suggestion. Indeed, we also considered the calculation of S1P/ApoM ratio and determination of its correlations with the studied inflammatory and lipid parameters. We could not find significant differences in S1P/ApoM ratio between FH patients and controls (2120±700 vs. 2290±740; p=0.22), moreover, there were marked inter-individual differences. The S1P/ApoM ratio showed significant negative correlation with sVCAM-1 (r=-0.197; p=0.037), while there was a positive correlation between S1P/ApoM and sCD40L (r=0.28; p=0.0027). We also added these data to the Results section (Table 1., Ln 117-119 and Ln 152-155).
However, because of the above-mentioned methodological issues, we did not used the S1P/ApoM ratio as the complex form, since based on this clinical study we cannot be sure that the ratio represents the ApoM associated S1P content. Therefore, we plan further studies to clarify the raised questions.
- How about the role of Lysophosphatidic acid (LPA) in heterozygous familial hypercholesterolemia.
Response: Thank you for this forward-looking comment. Lysophosphatidic acid (LPA; 1‐acyl 2‐hydroxyl glycerol 3‐phosphate) is a bioactive lipid component with multiple biological functions by regulating a variety of diseases. A former study reported higher serum levels of LPA in hypercholesterolemic rabbits (Tokumura A et al. J Lipid Res. 2002 Feb;43(2):307-15. DOI: 10.1016/S0022-2275(20)30173-5). Furthermore, LDL-associated LPA was increased in plasma from high-fat Western diet-fed mice that are genetically prone to hyperlipidemia. LDL from human subjects contained LPA and LPA content was decreased by lipid-lowering drug therapies. (Kraemer MP et al. J Lipid Res. 2019 Nov; 60(11): 1818–1828. doi: 10.1194/jlr.M093096).
To date, the best of my knowledge, the role of LPA in heterozygous familial hypercholesterolemia has not been studied. Therefore, the possible interactions between the ApoM/S1P axis and LPA regulation pathways are unexplored.
Again, we are very thankful for your valuable and thorough review.